# An Exploration of Biophilic Design Features within Preschool Interiors

**Ibtisam Mohammed [1,\*], Zeynep Onur [2] and Çiğdem Çağnan [3]**

1  Department of Architecture, Faculty of Engineering, University of Duhok, Duhok 42001, Iraq
2  Department of Architecture, Faculty of Architecture and Fine Arts, International Final University, Girne 99320, Cyprus; zeynep.onur@final.edu.tr
3  Department of Architecture, Faculty of Architecture, Near East University, Nicosia 99138, Cyprus; cigdem.cagnan@neu.edu.tr
\*  Correspondence: ibtsam_11@yahoo.com

**Abstract:** This study examined the incorporation of biophilic elements in Duhok preschool interiors and integrated them into the Biophilic Interior Design Matrix (BID-M). This approach holds great promise for the development of restorative environments. The Biophilic Design Matrix (BDM) is based on Kellert's list of biophilic design attributes, adjusted to suit preschool interiors. Photos and plan drawings were employed as methods to analyze 59 interior spaces within six preschools, both private and public. The biophilic matrix revealed the presence of 30% of biophilic attributes in the analyzed spaces, with an average score of 16.45 out of 53 total points. Individual scores ranged from 8 to 22 for each space, highlighting variations in biophilic design within the six preschools. Notably, the study identified a lack of biophilic features that foster human–nature relationships in the majority of preschools, albeit minimally observed in some spaces. Conversely, the environmental features scored higher than the average. This research highlights the importance of enhancing biophilic design elements, especially those that strengthen human–nature relationships. The results enhance biophilic design in educational settings, promoting nature-centered, restorative environments for future interventions. Furthermore, we propose an innovative approach for assessing and incorporating biophilia in interiors, recognizing its positive impact on human health and wellbeing.

**Keywords:** biophilia; preschool interiors; biophilic interior design matrix; biophilic design

## 1. Introduction

Modern living has made nature more and more difficult for individuals to access, but the yearning to connect with nature still exists. Children are particularly affected by this, as they have to stay indoors while separated from their parents, which is especially true for them. This lack of connection with nature is concerning, as studies have shown that there is an inherent need for humans to be connected with nature, known as biophilia. Biophilia is defined as an inherent need to be connected with nature [1]. Biophilia is a natural need to connect with nature, which has become essential for children's cognitive functioning and wellbeing [1–3]. Biophilia is not just a desire but a natural need that has been found to be essential for children's cognitive functioning and wellbeing. Including nature in urban environments created for children is one possible solution to address this issue. Biophilic design strategies require consideration of the people using the place, its location, and its function. Plants are one way to incorporate nature into space, but there are other ways to incorporate nature [4,5]. To aid interior designers, Kellert [6] proposed six biophilic elements that could be implemented in a given space. To detect and evaluate biophilic elements, a reliable technique is required. The six elements include environmental features, natural shapes and forms, natural patterns and processes, light and space, place-based relationships, and human–nature relationships [7].

A few studies have been conducted to create a trustworthy coding method for determining biophilic contents in various interior environments. Based on Kellert [6], the matrix has 6 biophilic design elements and 52 biophilic features. Twenty of the original seventy-two features were left out of the matrix. This was due to their inability to be visually analyzed and their lack of connection to the interior space [7]. The BID-M was created by McGee and Marshall-Baker [8] in order to evaluate the impact of biophilic features on playrooms in child healthcare facilities on wellbeing [7]. Because evidence-based design has been more popular as an interdisciplinary method of gathering and sharing knowledge since the beginning of the 1980s, designers have had more and more reasons to incorporate biophilic design into interiors [9]. As more research is conducted on the individual attributes and wider applications of the matrix scoring are made to show greater trends and research opportunities, this initial quantification of biophilia will continue to expand. Additionally, there is a need to find more evidence-based arguments in favor of integrating biophilia into interior spaces like play areas. Future research is necessary to establish the best applications for each feature. The development and testing of numerous attribute-friendly features of design could potentially provide insight into the benefits of biophilia and how to best use it in interiors.

In light of the above, this study aims to address the following research questions:

1. What is the average presence of biophilic attributes in the analyzed spaces, and how does this align with the overall score and percentage of biophilic attributes' presence?
2. How can the Biophilic Interior Design Matrix (BID-M) be used to identify biophilic design features in preschool interiors?
3. What is the extent of biophilic element incorporation within the interiors of Duhok preschools, based on the six biophilic design elements proposed by Kellert?

These research questions will guide the exploration of biophilic design features within the interiors of Duhok preschools, utilizing the Biophilic Interior Design Matrix (BID-M) and Kellert's proposed biophilic design elements as foundational frameworks. By addressing these research questions, this study aims to contribute to the understanding and application of biophilic design principles in preschool interiors, paving the way for creating healthier and more nurturing environments for young children.

The study used a photographic methodology to document biophilic attributes for preschool interiors in Duhok, using the Biophilic Interior Design Matrix to address the study's objective. The BID-M assisted in quantifying the numerous biophilic design attributes present in preschool settings. The goal of the current research was to explore the existence of biophilic features in the interiors of Duhok preschools and include them in the BID-M based on the six elements proposed by Kellert [6].

Biophilic design, despite its potential benefits, remains an area of interior design expertise that has received relatively little attention. By addressing the problem of integrating biophilic design features in interior design, this research contributes to bridging this gap and sheds light on the potential of utilizing the BID-M instrument for the identification and implementation of biophilic design elements [9].

By examining and incorporating biophilic features into the BID-M, this study aims to provide valuable insights into the biophilic attributes present in preschool interiors, highlighting their significance in promoting a healthier and more nurturing environment for young children. The findings of this research will contribute to the understanding and application of biophilic design principles in the context of preschool interiors, ultimately aiming to enhance the wellbeing and cognitive development of children.

Moving forward, this research serves as a foundation for further exploration and integration of biophilic design elements in interior design practice. By recognizing the potential of the BID-M as a tool for biophilic design assessment, designers and practitioners can more effectively incorporate nature-inspired elements into interior spaces, creating restorative and enriching environments for users.

## 2. Literature Review

### 2.1. The Concept of Biophilic Design and Its Benefits

Biophilic design is an architectural and interior design concept that aims to bring humans closer to nature by integrating natural components into the built environment. The notion behind the concept is that humans have an inbuilt connection to nature and that exposure to natural elements may improve health and wellbeing [10]. This is to say that there is a connection between nature and human wellbeing.

Biophilic design has both health and financial benefits, and neglecting nature can lead to a loss of profit [11,12]. According to an argument made by Browning et al. [11] (p. 3), "incorporating nature into the built environment is not just a luxury but a sound economic investment in health and productivity, based on well-researched neurological and physiological evidence". Biophilic design can lead to increased healing, learning, property values, and productivity while decreasing absenteeism, anxiety, and incarceration expenses. It is important to value and safeguard the environment [6]. Restorative environmental design (RED) is a strategy that aims to minimize and mitigate adverse impacts on the natural environment and foster beneficial contact between people and nature in contemporary buildings and landscapes. Kellert [6] defined it as a low-environmental-impact approach. Interior environment designers play a significant role in the development of interior/exterior links and features that can improve user exposure to biophilia, which can have positive effects on health [13,14] and wellbeing [15,16]. Restorative environmental design connects sustainability and biophilia, generating globally conscious individuals who value nature and desire to return it to its ideal state. The addition of biophilic design and imitating natural settings is clearly the next step in the sustainable design movement [17]. However, there is insufficient support for best practices for developing the interior environments that people naturally choose [18]. Biophilic design is an approach to creating healthy, sustainable, and adaptable environments. It also has benefits such as improved air quality, reduced stress levels, and increased productivity, as highlighted by Cacique, Maria, and Sheng-Jung Ou [19]. The term "biophilia" refers to the emotional bonds that people develop with other living species. The organic or naturalistic component, which combines direct and symbolic aspects of nature, and the place-based or vernacular dimension, which stresses the integration of architecture and landscapes with the environment and culture of an area, are the two dimensions of biophilic design. This architectural method generates a sense of place and identity, which can increase people's loyalty and responsibility for their built environment and heritage.

### 2.2. Sustainability and Biophilia

The concept of sustainability centers on the idea of addressing present demands without risking future generations' capacity to meet their own needs. The close relationship between humans and the natural environment is an important part of sustainability. Biophilia is a technique that provides a sustainable design strategy that includes reconnecting humans with nature [20]. Sustainable design balances human needs with natural and cultural environments, whereas biophilic design examines how the environment affects our physiology and psychology. Sustainable architecture is created in harmony with the environment and natural resources. Biophilia and sustainability can be achieved by preserving natural and cultural environments and using resources in site and building design to encourage exploration. This knowledge can be used to modify space to enhance human experiences. Biophilic design addresses sustainability through tactile, emotional, and experiential approaches, creating buildings that are resource-saving, energy-efficient, and improve the socialization, productivity, health, and wellbeing of their occupants [21].

### 2.3. Biophilic Interior Design and Tools

Biophilic design is a novel approach to interior design that tries to incorporate natural aspects into the built environment. Designers may create environments that are not only visually beautiful but also give multiple advantages to the occupants by including

biophilic features such as plants, natural sunlight, and organic materials. These biophilic characteristics not only improve the aesthetic attractiveness of the interior space but also improve the physical and emotional wellbeing of those who live in these places. Plants are one biophilic component that may be introduced into interior spaces. Indoor plants have been proven to provide several advantages to people. Plants have been shown in studies to enhance air quality by lowering levels of carbon dioxide and other pollutants.

The fundamental dimensions of biophilic design can be categorized into six key elements: environmental features, natural shapes and forms, natural patterns and processes, light and space, place-based relationships, and evolved human–nature relationships. These six elements encompass over 70 specific design attributes that contribute to creating a stronger connection between humans and nature. A Yale professor has extensively categorized biophilia into these elements and identified 72 specific features within them. For instance, within the environmental features element, air is recognized as one of its attributes. This comprehensive list of biophilic design attributes in landscape and architecture was developed based on the extensive exposure of the professor to a diverse range of authors and researchers in the field [6,18]. The Terrapin Bright Green List of 14 Patterns of Biophilic Design is an alternative tool that emphasizes recognizable patterns, nature–health linkages, and nature–design relationships [22]. Kellert's original list of 72 attributes is limited by a shorter list of 14 attributes. Leadership in Energy and Environmental Design (LEED) has raised awareness of sustainable design but has not provided holistic design strategies for biophilic design. Innovators in biophilic design, such as WELL Building and the Living Building Challenge, reference Kellert's original list of design attributes. Kellert's vocabulary for interior designers was extended further, allowing them to assist in WELL Building and the Living Building Challenge [18].

### 2.4. Identifying Biophilic Features in Interior Spaces

Color, light, and materiality are all connected design elements that occur together in an interior environment. Nevertheless, it is unknown how these three features are being applied in relation to biophilic features and how existing research supports biophilic features [23]. McGee and Marshall-Baker created the BDM to identify biophilic features in 24 children's healthcare interiors. A study in South Atlantic state Child Life play areas found consistent biophilic attributes in some settings while lacking in others [8]. The BDM serves as a valuable instrument for evaluating the presence and effectiveness of biophilic design in healthcare environments for children.

Marte et al. tested the BID-M in urban playrooms by coding photographs from 45 children's playrooms in Manhattan residential buildings. They evaluated overall and component inter-rater reliability, finding high overall reliability but low reliability in some components [7]. Designers' perceptions of biophilic design were explored by developing an updated Biophilic Interior Design Matrix (BDM) as a design tool. The results showed that practitioners had a better understanding of biophilic design, enabling designers to incorporate nature-based features in indoor settings [18].

McGee and Park examined biophilic interior design utilizing the BID-M vocabulary and its essential elements. They found that practitioners incorporated a range of biophilic attributes into their work, with color preferences being the top attribute. Abstraction of nature, composition, natural light, and natural materials were the top attributes that practitioners and the literature review had in common. Additional research is needed to fully understand how biophilic design can be variedly included for the best natural interior settings [23]. McGee et al. evaluated how Chinese design professionals perceived biophilia and the BID-M, as well as their reliability and validity. A pre- and post-questionnaire was used with 101 interior designers and architects, and the Chinese designer's translation of the BID-M was seen as reliable, valid, and helpful in the design process [24].

### 2.5. The Importance of Biophilic Design in Preschool Interiors

The use of natural features such as plants, natural materials, and lighting is an essential part of biophilic design in preschool settings. Awad et al. investigated the significance of incorporating biophilic design principles into elementary school design regulations, such as lighting, daylight, materials, and ventilation, to enhance the wellbeing and health of children [25]. Meanwhile, Ondul et al. aimed to design a "Refuge Pod" piece of furniture using a biophilic design approach in a preschool setting while taking the demands of the space into consideration for both furniture and space. The research revealed that children are imaginative and like to play in tiny areas [26], underscoring the significance of establishing personal and engaging places.

In a study conducted by Yanez et al., young children from urban and rural areas were compared in terms of their biophilia and attitudes towards nature. The results revealed no significant variation based on geographical region. The study identified common themes, including the definition of nature, awareness of environmental effects, and adherence to natural rules. The researchers suggested that cognitive development and personal preferences might be more accurate indicators in this context [27]. In another study by Ünal and Özen, the benefits of exposing preschool children to nature through biophilic design in built environments were investigated. The researchers used content analysis, cognitive mapping, and semi-structured interviews to assess the overall score. They found that a statistically significant impact was observed when the *p*-value was less than 0.050 [28]. Showing a positive impact through contact with nature, Yassein and Ebrahiem discussed contemporary biophilic interior design techniques used in residential and workplace settings based on a discussion with architecture students. This situated the notion of biophilic interior design and resulted in a conceptual model that strengthens and supports the human–nature relationship [29]. This further reinforces the connection between humans and nature.

### 2.6. The Impact of Biophilic Design on Children's Learning and Development

Incorporating biophilic design into educational facilities has various advantages, including higher test scores, greater health, and increased learning. The benefits extend to recreational spaces such as playgrounds, where biophilic design concepts can enhance children's behavior, attention span, and mental health. Biophilic design patterns have been demonstrated to improve stress reduction, cognitive abilities, sensitivity, atmosphere, and preference. According to the research findings, biophilic design boosts liking for the environment, decreases stress, increases happiness, and encourages focus [30]. McGee measured biophilic design attributes in Child Life settings to improve health and wellbeing. The study utilized a biophilic interior design matrix tool and an open-ended questionnaire to create 24 case studies. The results indicated that there is a correlation between biophilic environments and effective Child Life play areas, with higher scores representing greater effectiveness. Another study involved 90 Child Life specialists who used a photographic technique to identify elements that contribute to the design of ideal playrooms in different hospitals. The findings highlighted the crucial role of Child Life specialists in designing or modifying these settings [31,32]. Schools and educational institutions play a significant role in fostering a connection with nature. By providing a natural environment, these institutions can offer benefits to young people, such as promoting the use of nature, enhancing attention, cognitive performance, and mood. Moreover, a natural environment in schools improves concentration and self-discipline and reduces physiological stress [33]. The concept of biophilia might be used in the classroom to contribute to the creation of ecologically sound urban environments [20]. Fadda et al.'s research specifically demonstrated the beneficial effects of biophilic design on preschoolers' visual attention in indoor environments. Notably, even a brief exposure to plants on a desk for 48 seconds increased the duration of children's initial visual fixation on art displays [34].

There is a lack of research on the topic of integrating biophilic design elements into preschool interiors. By doing this, we can enhance children's cognitive abilities, lessen their

stress levels, and foster their creativity. Additionally, it can encourage them to preserve nature by fostering a closer connection with it. But the Biophilic Design Matrix (BDM) offers a chance to advance design research while enhancing the health, security, and happiness of users. In an effort to address the difficulty of quantitatively evaluating biophilia, the BDM was created in 2011, with a focus on children's and adolescents' play areas. The BDM was successful in providing a visual inventory of biophilic features and the frequency of their use in hospital spaces intended for patient recreation and play. The current state of biophilic design research and practice in this context has also been explored in further research.

## 3. Materials and Methods

Biophilic design is a recognized methodology used to evaluate and incorporate nature-inspired elements in architectural settings, acknowledging the inherent need of children to be in environments with biophilic design patterns for their wellbeing [35]. In line with this, the objective of this study was to develop the biophilic interior design matrix (BID-M), specifically tailored for assessing biophilic contents in preschool interiors, utilizing Kellert's well-established design attributes [6]. These attributes, proposed by Kellert, encompass six biophilic design elements, including environmental features, natural shapes and forms, natural patterns and processes, light and space, place-based relationships, and human–nature relationships.

The BDM, initially developed by McGee and Marshall-Baker for identifying biophilic design elements in pediatric hospitals' play environments, served as the foundation for this study [8]. Notably, this is the first application of the BDM as a tool for interior design. Out of Kellert's 72 biophilic design elements, including those related to landscape and architecture, 53 were deemed appropriate for interior design purposes, while the remaining 19 were considered unsuitable [18]. This is because the 19 excluded are considered to be more relevant to landscape and architecture.

Accordingly, the present study aimed to expand the potential applications of the biophilic interior design matrix in different types of settings within Duhok, with a particular focus on preschool interiors. The BID-M technique employs a quantitative approach to evaluate biophilic features, providing a rating for each space and describing the presence of attributes both verbally and photographically. A total of 59 interior spaces from six preschools, consisting of four public and two private institutions, were selected for analysis (see Tables 1 and 2). Prior to data collection, the research team obtained the necessary approval letter from the Directorate of Education to ensure access to the preschools and the collection of relevant information for the study.

**Table 1.** The biophilic design elements and attributes.

| | Environmental Features | | Light and Space |
|---|---|---|---|
| 1 | Color | 32 | Natural light |
| 2 | Water | 33 | Filtered and diffused light |
| 3 | Air | 34 | Light and shadow |
| 4 | Sunlight | 35 | Reflected light |
| 5 | Plants | 36 | Light pools |
| 6 | Animals | 37 | Warm light |
| 7 | Natural material | 38 | Light as shape and form |
| 8 | Views and vistas | 39 | Spaciousness |
| 9 | Fire | 40 | Spatial variability |
| | Natural Shapes and Forms | 41 | Space as shape and form |
| 10 | Botanical motifs | 42 | Spatial harmony |
| 11 | Tree and columnar supports | 43 | Inside/outside spaces |
| 12 | Animal | | Place-Based Relationships |

**Table 1.** *Cont.*

| | | | | |
|---|---|---|---|---|
| 13 | Shells and spirals (invertebrates) | 44 | Geographic connection to place |
| 14 | Egg, oval, and tubular forms | 45 | Historic connection to place |
| 15 | Arches, vaults, domes | 46 | Ecological connection to place |
| 16 | Shapes resisting straight lines | 47 | Cultural connection to place |
| 17 | Simulation of natural features/biomorphy | 48 | Indigenous materials |
| 18 | Geomorphology | 49 | Landscape orientation/landscape features |
| 19 | Biomimicry | | Human–Nature Relationships |
| | Natural Patterns and Processes | 50 | Prospect and refuge |
| 20 | Sensory variability/information richness | 51 | Order and complexity |
| 21 | Age, change, the patina of time | 52 | Curiosity and enticement |
| 22 | Central focal point | 53 | Change and metamorphosis |
| 23 | Patterned wholes | | |
| 24 | Bounded spaces | | |
| 25 | Transitional spaces | | |
| 26 | Linked series and chains | | |
| 27 | Integrations of parts to wholes | | |
| 28 | Complementary contrasts | | |
| 29 | Dynamic balance and tension | | |
| 30 | Fractals | | |
| 31 | Hierarchically organized ratios and scales | | |

**Table 2.** The biophilic interior design matrix scores of Chiman preschool settings.

| | Chiman Preschool | Multipurpose Hall | Cafeteria | Classroom 1 | Classroom 2 | Classroom 3 | Classroom 4 | Classroom 5 | Classroom 6 | Corridors | Total Subscores |
|---|---|---|---|---|---|---|---|---|---|---|---|
| | **(1) Environmental features** | | | | | | | | | | |
| 1 | Color | ✓ | ✓ | ✓ | ✓ | ✓ | ✓ | ✓ | ✓ | ✓ | 9 |
| 2 | Water | × | ✓ | × | × | × | × | × | × | × | 1 |
| 3 | Air | ✓ | ✓ | ✓ | ✓ | ✓ | ✓ | ✓ | ✓ | ✓ | 9 |
| 4 | Sunlight | ✓ | ✓ | ✓ | ✓ | ✓ | ✓ | ✓ | ✓ | ✓ | 9 |
| 5 | Plants | × | × | × | × | × | × | × | × | × | 0 |
| 6 | Animals | × | × | × | × | × | × | × | × | × | 0 |
| 7 | Natural material | × | × | × | × | × | × | × | × | × | 0 |
| 8 | Views and vistas | ✓ | ✓ | ✓ | ✓ | ✓ | ✓ | ✓ | ✓ | ✓ | 9 |
| 9 | Fire | × | × | × | × | × | × | × | × | × | 0 |
| | Subscores | 4 | 5 | 4 | 4 | 4 | 4 | 4 | 4 | 4 | 37 |
| | **(2) Natural shapes and forms** | | | | | | | | | | |
| 10 | Botanical motifs | × | ✓ | ✓ | ✓ | ✓ | ✓ | ✓ | ✓ | ✓ | 8 |
| 11 | Tree and columnar supports | × | ✓ | × | × | × | × | × | × | × | 1 |
| 12 | Animal | ✓ | ✓ | ✓ | ✓ | × | ✓ | × | ✓ | × | 6 |
| 13 | Shells and spirals (invertebrates) | × | × | × | × | ✓ | × | ✓ | × | × | 2 |

**Table 2.** *Cont.*

| Chiman Preschool | | Multipurpose Hall | Cafeteria | Classroom 1 | Classroom 2 | Classroom 3 | Classroom 4 | Classroom 5 | Classroom 6 | Corridors | Total Subscores |
|---|---|---|---|---|---|---|---|---|---|---|---|
| 14 | Egg, oval, and tubular forms | × | ✓ | × | × | × | × | × | × | × | 1 |
| 15 | Arches, vaults, domes | × | × | × | × | × | × | × | × | × | 0 |
| 16 | Shapes resisting straight lines | × | × | × | × | × | × | × | × | × | 0 |
| 17 | Simulation of natural features/biomorphy | × | × | × | × | × | × | × | × | × | 0 |
| 18 | Geomorphology | × | × | × | × | × | × | × | × | × | 0 |
| 19 | Biomimicry | × | × | × | × | × | × | × | × | × | 0 |
| | Subscores | 1 | 4 | 2 | 2 | 2 | 2 | 2 | 2 | 1 | 18 |
| **(3) Natural patterns and processes** | | | | | | | | | | | |
| 20 | Sensory variability/information richness | ✓ | ✓ | ✓ | ✓ | ✓ | ✓ | ✓ | ✓ | ✓ | 9 |
| 21 | Age, change, the patina of time | × | × | × | × | × | × | × | × | × | 0 |
| 22 | Central focal point | × | ✓ | × | × | × | × | × | × | × | 1 |
| 23 | Patterned wholes | × | × | × | × | × | × | × | × | × | 0 |
| 24 | Bounded spaces | ✓ | × | ✓ | ✓ | ✓ | ✓ | ✓ | ✓ | × | 7 |
| 25 | Transitional spaces | × | × | × | × | × | × | × | × | ✓ | 1 |
| 26 | Linked series and chains | × | ✓ | × | × | × | × | × | × | ✓ | 2 |
| 27 | Integrations of parts to wholes | ✓ | ✓ | ✓ | ✓ | ✓ | ✓ | ✓ | ✓ | ✓ | 9 |
| 28 | Complementary contrasts | × | × | × | × | × | × | × | × | × | 0 |
| 29 | Dynamic balance and tension | × | × | × | × | × | × | × | × | × | 0 |
| 30 | Fractals | × | × | × | × | × | × | × | × | × | 0 |
| 31 | Hierarchically organized ratios and scales | × | × | × | × | × | × | × | × | × | 0 |
| | Subscores | 3 | 4 | 3 | 3 | 3 | 3 | 3 | 3 | 4 | 29 |
| **(4) Light and space** | | | | | | | | | | | |
| 32 | Natural light | ✓ | ✓ | ✓ | ✓ | ✓ | ✓ | ✓ | ✓ | ✓ | 9 |
| 33 | Filtered and diffused light | ✓ | ✓ | ✓ | ✓ | ✓ | ✓ | ✓ | ✓ | ✓ | 9 |
| 34 | Light and shadow | ✓ | ✓ | ✓ | ✓ | ✓ | ✓ | ✓ | ✓ | ✓ | 9 |
| 35 | Reflected light | × | × | × | × | × | × | × | × | × | 0 |
| 36 | Light pools | × | × | × | × | × | × | × | × | × | 0 |
| 37 | Warm light | × | × | × | × | × | × | × | × | × | 0 |
| 38 | Light as shape and form | × | ✓ | × | × | × | × | × | × | × | 1 |
| 39 | Spaciousness | ✓ | ✓ | ✓ | ✓ | ✓ | ✓ | ✓ | ✓ | × | 8 |
| 40 | Spatial variability | × | ✓ | × | × | × | × | × | × | × | 1 |
| 41 | Space as shape and form | × | × | × | × | × | × | × | × | × | 0 |
| 42 | Spatial harmony | × | × | × | × | × | × | × | × | × | 0 |
| 43 | Inside/outside spaces | × | ✓ | × | × | × | × | × | × | ✓ | 2 |
| | Subscores | 4 | 7 | 4 | 4 | 4 | 4 | 4 | 4 | 4 | 39 |

| | Chiman Preschool | Multipurpose Hall | Cafeteria | Classroom 1 | Classroom 2 | Classroom 3 | Classroom 4 | Classroom 5 | Classroom 6 | Corridors | Total Subscores |
|---|---|---|---|---|---|---|---|---|---|---|---|
| | (5) Place-based relationships | | | | | | | | | | |
| 44 | Geographic connection to place | ✓ | ✓ | ✓ | ✓ | ✓ | ✓ | ✓ | ✓ | ✓ | 9 |
| 45 | Historic connection to place | × | × | × | × | × | × | × | × | × | 0 |
| 46 | Ecological connection to place | ✓ | × | × | × | × | × | × | × | × | 1 |
| 47 | Cultural connection to place | ✓ | ✓ | ✓ | ✓ | ✓ | ✓ | ✓ | ✓ | ✓ | 9 |
| 48 | Indigenous materials | × | × | × | × | × | × | × | × | × | 0 |
| 49 | Landscape orientation/landscape features | ✓ | × | ✓ | ✓ | ✓ | ✓ | ✓ | ✓ | ✓ | 8 |
| | Subscores | 4 | 2 | 3 | 3 | 3 | 3 | 3 | 3 | 3 | 27 |
| | (6) Human–nature relationships | | | | | | | | | | |
| 50 | Prospect and refuge | × | × | × | × | × | × | × | × | × | 0 |
| 51 | Order and complexity | × | × | × | × | × | × | × | × | × | 0 |
| 52 | Curiosity and enticement | × | × | × | × | × | × | × | × | × | 0 |
| 53 | Change and metamorphosis | × | × | × | × | × | × | × | × | × | 0 |
| | Subscores | 0 | 0 | 0 | 0 | 0 | 0 | 0 | 0 | 0 | 0 |
| | Total subscores | 16 | 22 | 16 | 16 | 16 | 16 | 16 | 16 | 16 | 150 |
| | Average | | | | | | | | | | 16.66 |

In summary, the materials and methods of this study involved the development and application of the Biophilic Interior Design Matrix (BID-M), specifically tailored for preschool interiors in Duhok. The BID-M incorporates Kellert's design attributes and was used quantitatively to rate and describe the biophilic contents of each space within the selected preschools. Ethical considerations were taken into account, and approval was obtained from the Directorate of Education to conduct the research in the chosen preschools.

*Methods*

Purposive sampling was used to pick cases (preschools) in this research study. The researchers purposefully chose 6 preschools from a total of 21 accessible in Duhok that would best fit the research objectives. Purposeful sampling is a non-probability sampling strategy in which cases or participants are intentionally chosen based on certain qualities or attributes important to the research [36,37]. To guarantee variety in the sample, the researchers carefully picked preschools that represented several types of settings, including both public and private institutions. The researchers were able to guarantee that the selected instances were typical of the target population and could provide significant data for analysis by carefully selecting preschools that might provide important insights and information connected to the study questions. The researchers were able to focus on certain elements that were critical for the study, such as a variety of interior locations within the preschools, according to deliberate sampling. This systematic approach to case selection enabled a comprehensive assessment of biophilic design features in preschool environments, since the chosen cases provided the essential variance and depth of understanding.

The study was able to purposefully select preschools that reflected the required diversity and qualities essential to successfully examine and comprehend biophilic design aspects in preschool interiors by using purposeful sampling in this research. The researchers

analyzed each location within the preschools using observation and photography, including the ceiling, walls, floor, and furniture. The objective was to see whether there were any special biophilic features in these places. This evaluation was led by Kellert's framework's six biophilic design principles and their related qualities. The researchers were able to analyze and score each space based on the identified biophilic features through meticulous observation and photographic recording. The visual attributes and features of the rooms were examined to see whether they displayed biophilic design principles. The results of this evaluation were utilized to build the Biophilic Interior Design Matrix, which provided insights into the presence and distribution of biophilic features throughout the preschools' various interior areas.

The use of observation and photography enabled a thorough analysis of the physical elements of the interior spaces. This method offered a visual record and documentation of the presence or absence of biophilic features within each space (see Figure 1), allowing for a more objective evaluation of the biophilic design elements present in the preschool environments.

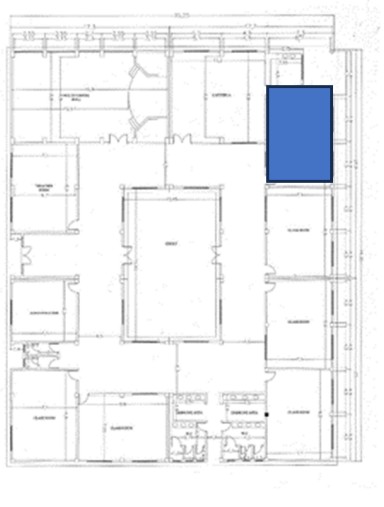

(**a**)

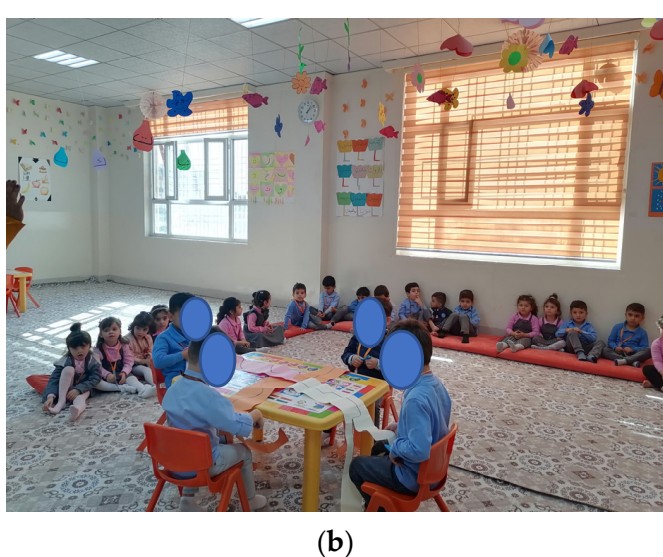

(**b**)

**Figure 1.** Class room of Shang preschool: (**a**) plan drawing from the directorate of the preschool; (**b**) photo by the author.

The biophilic design attributes of each preschool's spaces were determined separately by creating a matrix that included the number of spaces that would be allocated along the columns on the one hand and the set of biophilic features along the rows on the other. By indicating with a (✓) sign if present and a (×) sign if not present, these signs were converted to numbers (✓ = 1 and × = 0), in order to evaluate all of the listed attributes in a specific space. The maximum score for various biophilic features in the matrix was 53. In order to determine the overall biophilic inclusion of any space within the preschool, the biophilic attributes of each of the six biophilic elements were combined to create a subscore, as shown in Tables 1 and 2.

## 4. Results and Discussion

The quantitative analysis of the matrix was conducted to assess and compare the biophilic attributes among the preschools in Duhok. Scoring in the matrix was based on the presence or absence of specific attributes, enabling an evaluation of biophilia. The average overall score of the 59 analyzed interiors was determined to be 16.45. This score was then divided by the total number of features (53), resulting in a calculated percentage of 30% for the extent of biophilic presence, as presented in Table 3. Each attribute in the matrix contributed one score towards the assessment. The recorded scores ranged from 8 to 22, indicating the range of variation in the level of biophilic incorporation within each space.

These findings are visually represented and organized in Figure 2, offering a graphical depiction of the distribution of biophilic attributes across the analyzed preschool interiors.

**Table 3.** The biophilic interior design matrices of the whole preschools.

| | Biophilic Interior Design Matrix | Shang Preschool | Daveen Preschool | Avrocity Preschool | Zaryland Preschool | Kapir Preschool | Cihman Preschool | Total Scores | Average |
|---|---|---|---|---|---|---|---|---|---|
| | (1) Environmental features | | | | | | | | |
| 1 | Color | 9 | 9 | 13 | 10 | 9 | 9 | 59 | |
| 2 | Water | 2 | 1 | 12 | 1 | 1 | 1 | 18 | |
| 3 | Air | 9 | 9 | 13 | 8 | 9 | 9 | 57 | |
| 4 | Sunlight | 9 | 9 | 13 | 8 | 9 | 9 | 57 | |
| 5 | Plants | 1 | 0 | 0 | 0 | 1 | 0 | 2 | |
| 6 | Animals | 0 | 0 | 0 | 0 | 0 | 0 | 0 | |
| 7 | Natural material | 0 | 0 | 0 | 0 | 2 | 0 | 2 | |
| 8 | Views and vistas | 9 | 9 | 12 | 8 | 9 | 9 | 56 | |
| 9 | Fire | 0 | 0 | 0 | 0 | 0 | 0 | 0 | |
| | Subscores | 39 | 37 | 63 | 35 | 40 | 37 | 251 | 4.25 |
| | (2) Natural shapes and forms | | | | | | | | |
| 10 | Botanical motifs | 8 | 8 | 13 | 7 | 7 | 8 | 51 | |
| 11 | Tree and columnar supports | 0 | 1 | 0 | 0 | 0 | 1 | 2 | |
| 12 | Animal | 2 | 9 | 11 | 4 | 5 | 6 | 37 | |
| 13 | Shells and spirals (invertebrates) | 6 | 0 | 0 | 3 | 0 | 2 | 11 | |
| 14 | Egg, oval, and tubular forms | 0 | 1 | 0 | 0 | 0 | 1 | 2 | |
| 15 | Arches, vaults, domes | 0 | 0 | 9 | 0 | 0 | 0 | 9 | |
| 16 | Shapes resisting straight lines | 0 | 0 | 9 | 0 | 0 | 0 | 9 | |
| 17 | Simulation of natural features/biomorphy | 0 | 0 | 0 | 0 | 0 | 0 | 0 | |
| 18 | Geomorphology | 0 | 0 | 0 | 0 | 0 | 0 | 0 | |
| 19 | Biomimicry | 0 | 0 | 0 | 0 | 0 | 0 | 0 | |
| | Subscores | 16 | 19 | 42 | 14 | 12 | 18 | 121 | 2.05 |
| | (3) Natural patterns and processes | | | | | | | | |
| 20 | Sensory variability/information richness | 9 | 9 | 13 | 10 | 9 | 9 | 59 | |
| 21 | Age, change, the patina of time | 0 | 0 | 0 | 0 | 0 | 0 | 0 | |
| 22 | Central focal point | 1 | 1 | 1 | 0 | 1 | 1 | 5 | |
| 23 | Patterned wholes | 0 | 0 | 0 | 0 | 0 | 0 | 0 | |
| 24 | Bounded spaces | 9 | 7 | 13 | 10 | 9 | 7 | 55 | |
| 25 | Transitional spaces | 1 | 1 | 0 | 1 | 1 | 1 | 5 | |
| 26 | Linked series and chains | 1 | 2 | 1 | 1 | 1 | 2 | 8 | |
| 27 | Integrations of parts to wholes | 9 | 9 | 13 | 10 | 9 | 9 | 59 | |
| 28 | Complementary contrasts | 0 | 0 | 11 | 0 | 0 | 0 | 11 | |
| 29 | Dynamic balance and tension | 0 | 0 | 0 | 0 | 0 | 0 | 0 | |

**Table 3.** *Cont.*

| | Biophilic Interior Design Matrix | Shang Preschool | Daveen Preschool | Avrocity Preschool | Zaryland Preschool | Kapir Preschool | Cihman Preschool | Total Scores | Average |
|---|---|---|---|---|---|---|---|---|---|
| 30 | Fractals | 0 | 0 | 0 | 0 | 0 | 0 | 0 | |
| 31 | Hierarchically organized ratios and scales | 0 | 0 | 0 | 0 | 0 | 0 | 0 | |
| | Subscores | 30 | 29 | 52 | 32 | 30 | 29 | 202 | 3.42 |
| | (4) Light and space | | | | | | | | |
| 32 | Natural light | 9 | 9 | 13 | 9 | 9 | 9 | 58 | |
| 33 | Filtered and diffused light | 9 | 9 | 12 | 9 | 9 | 9 | 57 | |
| 34 | Light and shadow | 9 | 9 | 13 | 9 | 9 | 9 | 58 | |
| 35 | Reflected light | 0 | 0 | 0 | 0 | 0 | 0 | 0 | |
| 36 | Light pools | 0 | 0 | 0 | 0 | 0 | 0 | 0 | |
| 37 | Warm light | 0 | 0 | 0 | 0 | 0 | 0 | 0 | |
| 38 | Light as shape and form | 0 | 1 | 1 | 0 | 0 | 1 | 3 | |
| 39 | Spaciousness | 8 | 7 | 1 | 0 | 1 | 8 | 25 | |
| 40 | Spatial variability | 1 | 7 | 1 | 0 | 1 | 1 | 11 | |
| 41 | Space as shape and form | 0 | 0 | 0 | 0 | 0 | 0 | 0 | |
| 42 | Spatial harmony | 0 | 0 | 0 | 0 | 0 | 0 | 0 | |
| 43 | Inside/outside spaces | 0 | 1 | 1 | 0 | 0 | 2 | 4 | |
| | Subscores | 36 | 43 | 42 | 27 | 29 | 39 | 216 | 3.66 |
| | (5) Place-based relationships | | | | | | | | |
| 44 | Geographic connection to place | 9 | 9 | 13 | 10 | 9 | 9 | 59 | |
| 45 | Historic connection to place | 0 | 0 | 0 | 0 | 0 | 0 | 0 | |
| 46 | Ecological connection to place | 1 | 1 | 13 | 0 | 0 | 1 | 16 | |
| 47 | Cultural connection to place | 9 | 9 | 13 | 10 | 9 | 9 | 59 | |
| 48 | Indigenous materials | 0 | 0 | 0 | 0 | 0 | 0 | 0 | |
| 49 | Landscape orientation/landscape features | 8 | 9 | 0 | 0 | 7 | 8 | 32 | |
| | Subscores | 27 | 28 | 39 | 20 | 25 | 27 | 166 | 2.81 |
| | (6) Human–nature relationships | | | | | | | | |
| 50 | Prospect and refuge | 1 | 0 | 11 | 1 | 2 | 0 | 15 | |
| 51 | Order and complexity | 0 | 0 | 0 | 0 | 0 | 0 | 0 | |
| 52 | Curiosity and enticement | 0 | 0 | 0 | 0 | 0 | 0 | 0 | |
| 53 | Change and metamorphosis | 0 | 0 | 0 | 0 | 0 | 0 | 0 | |
| | Subscores | 1 | 0 | 11 | 1 | 2 | 0 | 15 | 0.25 |
| | Total subscores | 149 | 156 | 249 | 129 | 138 | 150 | 971 | |
| | Average | 16.55 | 17.33 | 19.15 | 12.9 | 15.33 | 16.66 | 16.45 | 16.45 |

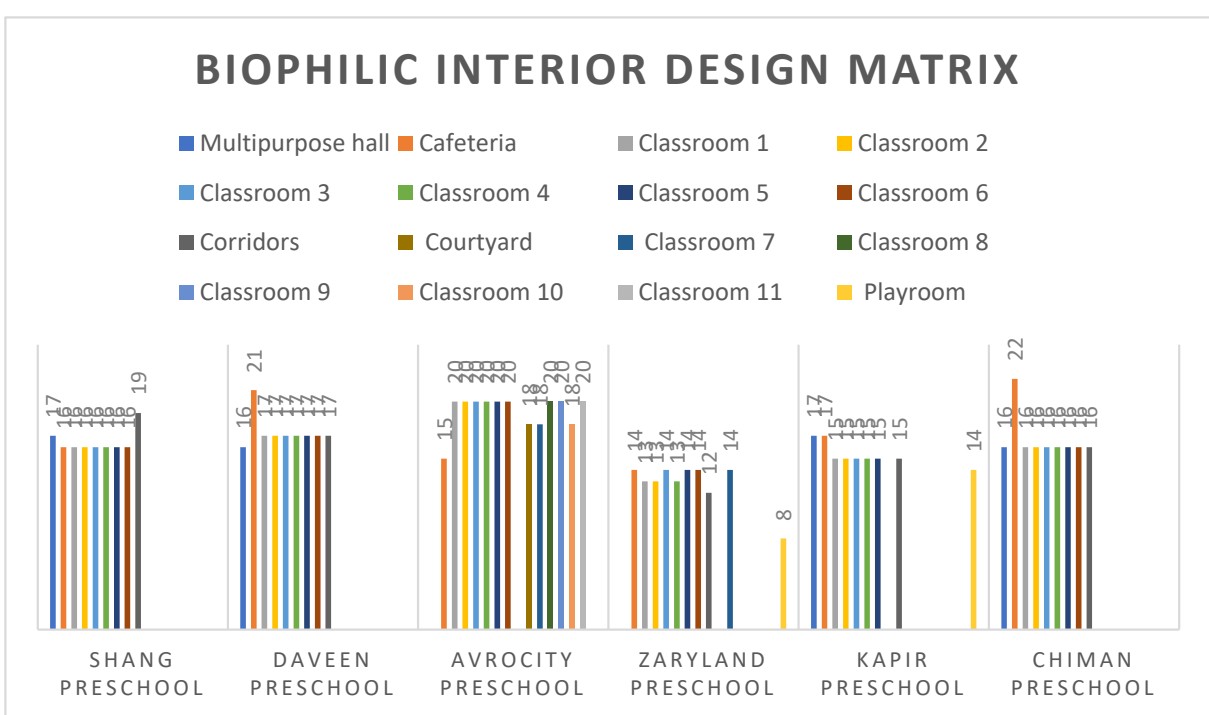

**Figure 2.** The biophilic interior design matrix of each space within the preschool.

The biophilic matrix of each preschool was organized into a separate table. Each table includes a set of preschool spaces distributed along the columns and a set of biophilic features distributed along the rows. Every space has subscores resulting from each existing attribute. These subscores were collected and the averages extracted to compare with one another in order to find the variances from the total average. The matrix results revealed that Avrocity preschool settings received a score of 19.15, higher than the average of the biophilic design of other preschools. The classrooms rated higher than average, while the cafeteria rated lower (Figures 2–4). The highest scoring was indicated through environmental features (4.25), followed by natural patterns and processes (3.42), as shown in Table 3.

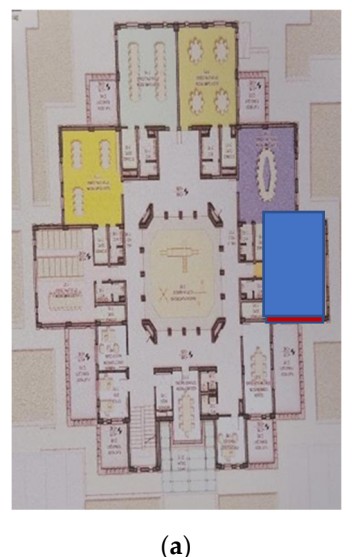

(**a**)

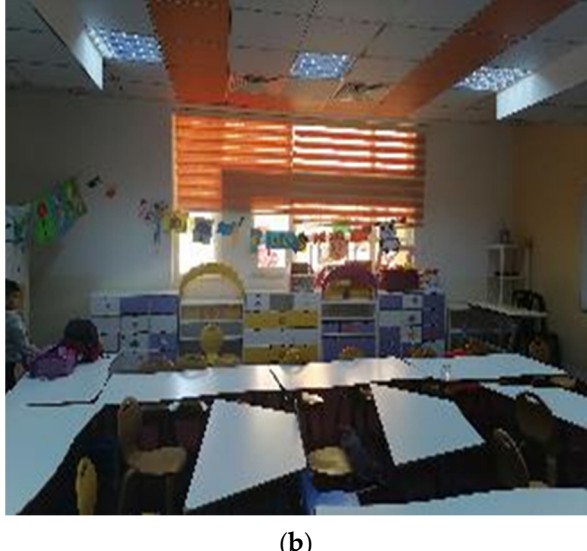

(**b**)

**Figure 3.** Classroom of Avrocity preschool: (**a**) plan drawing from the directorate of the preschool; (**b**) photo by the author.

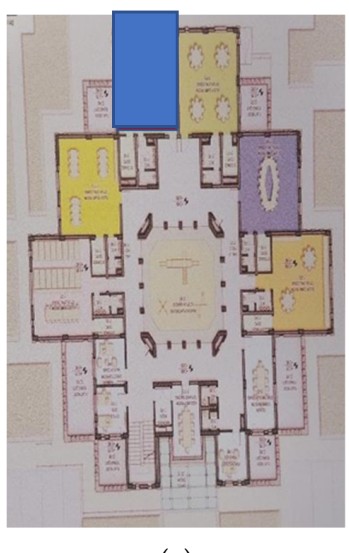
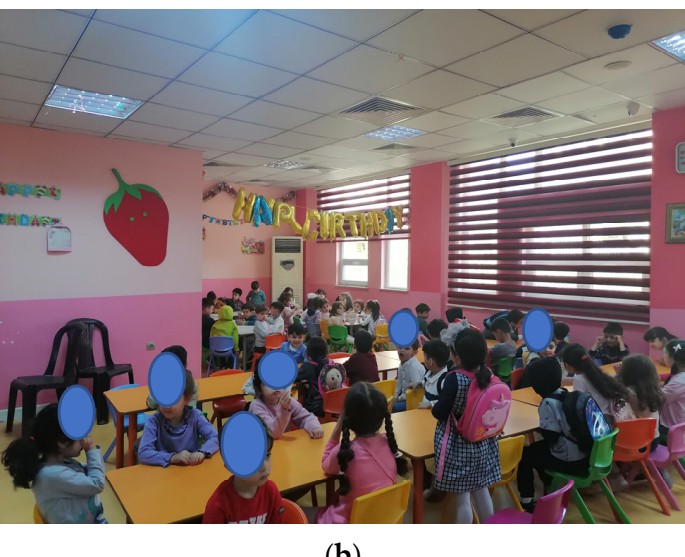

(**a**)                                                                    (**b**)

**Figure 4.** Cafeteria of Avrocity preschool: (**a**) plan drawing from the directorate of the preschool; (**b**) photo by the author.

The next highest score of 17.33 was indicated in Daveen preschool settings through the light and space elements. The Chiman and Shang preschool spaces received scores of 16.66 and 16.55, respectively, which were nearly equal to the average of the biophilic design in the matrices. Kapir preschool spaces received a score of 15.33, below the average score, and Zaryland preschool settings scored 12.9 for biophilic design.

The highest scoring was presented through the environmental features, while the lowest was indicated through the human–nature relationships element. The cafeteria of Chiman preschool scored the highest in biophilia. The space was large and spacious and also provided direct and indirect biophilic features like color, water, air, and views. The space offered botanical motifs, animal representations, and a tubular form of column. The height of the celling and the skylight provided a central focal point for the space, as well as natural light (see Figure 5).

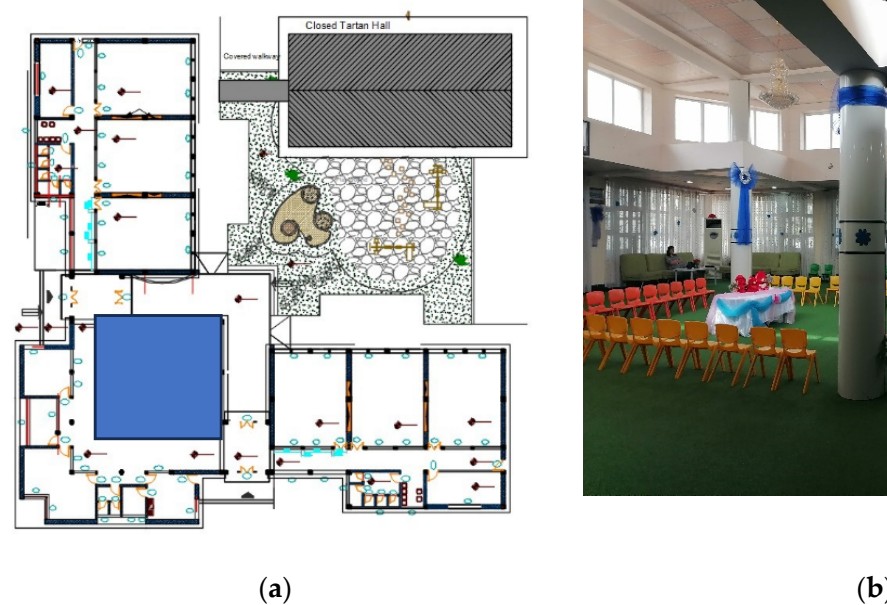
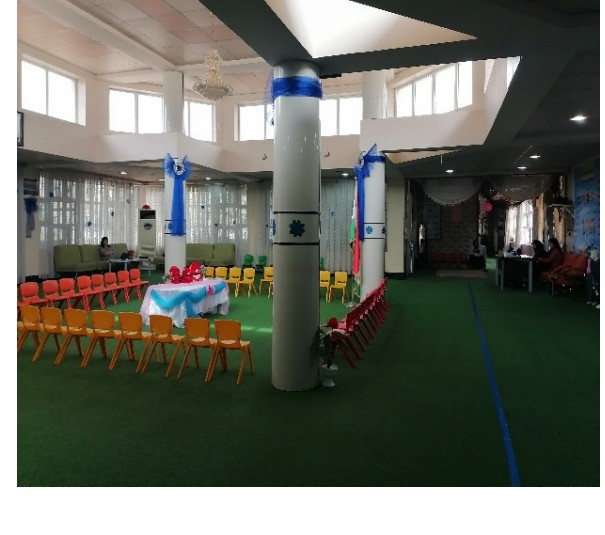

(**a**)                                                                    (**b**)

**Figure 5.** Cafeteria of Chiman preschool: (**a**) plan drawing from the directorate of the preschool; (**b**) photo by the author.

The playroom in Zaryland preschool scored the lowest rating for biophilic design attributes in the matrix. Although it was a small space without windows, it offered the absence of natural light, air, and views. There are opportunities to improve the room if some adjustments could be made, such as expanding the area or adding a window. The ceiling adds to the space and the integration of parts into the whole, where each piece, when joined together, creates something greater. As well as adding sensory variability and information richness through various colors, toys added visual variety to the space (see Figure 6).

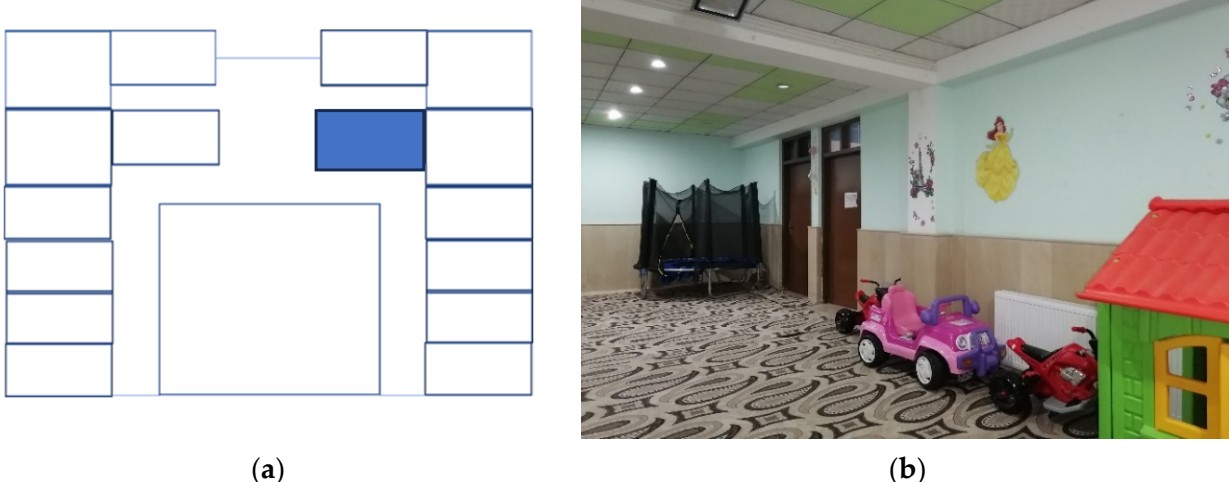

(**a**)            (**b**)

**Figure 6.** Playroom of Zaryland preschool: (**a**) plan drawing from the directorate of the preschool; (**b**) photo by the author.

The environmental features element scored higher than the average, such as color, which could be found in all spaces. This element is classified as the organic dimension, which offers direct features (e.g., air, sunlight) and indirect features (e.g., views and vistas) that can be seen in most spaces. The highest ratings of the attributes among all elements were found in color, sensory variability, and information richness (e.g., sound from gaming devices, laptops, and TV), or various colors, which have sensory diversity, integration of parts into wholes (e.g., ceiling patterns), and geographical and cultural connections to place (e.g., near mountains, natural vegetation). The second-highest attributes were natural light and light and shadow (e.g., windows, skylights). Additionally, following in the sequence were some features such as air, sunlight, filtered and diffused light, and views and vistas that were present in most of the spaces. In contrast, the natural shapes and forms and human–nature relationships elements both displayed lower scores than the average, as indicated in Table 3. This may occasionally happen because designing a space with some direct and indirect biophilic features may be easier than incorporating another biophilic attribute, which may seem more challenging.

The features that received the lowest rankings among all elements included order and complexity, curiosity and enticement, change and metamorphosis, historic connection to place, indigenous materials, space as shape and form, spatial harmony, reflected light, light pools, warm light, age, change, the patina of time, patterned wholes, dynamic balance and tension, fractals, hierarchically organized ratios and scales, geomorphology, biomimicry, simulation of natural features, animals, and fire (fire is a common issue, particularly in preschool settings, and should be considered in safe spaces), as indicated in Table 3. None of these features were observed in any observed space. While more research may justify increasing features for safe, proper incorporation in children's spaces, the presence of biophilia was increased and supported by design elements with multiple attributes in the children's space. By integrating various attributes and undertaking additional research to determine their impact, this can assist in the development of future products [31]. It has

been found that biophilia's engagement style in spaces is generally passive, whereas active engagement necessitates immersive experiences in nature.

### 4.1. Case Studies

Six preschools, both public and private, were selected to be analyzed in this study. These six preschools were selected from among many existing preschools within Duhok, due to the difficulty of obtaining permission to visit and photograph them for the collection of data required to conduct this research. As a result, 59 spaces were able to be included in the study.

#### 4.1.1. Shang Preschool

One of the typical public preschools included nine spaces to be analyzed: six classrooms, a multipurpose hall, a cafeteria, and corridors. All spaces shared the presence of the same direct biophilic features (i.e., fresh air, sunlight) and indirect features (i.e., various colors, natural views for vegetation) through operable windows or doors. Additionally, the cafeteria and corridors added water (a sink) as another indirect biophilic feature. Botanical motifs, representations, or symbolic themes in paintings were found in most of the spaces, except for the multipurpose hall. In addition, animal representation features were found only in the multipurpose hall and corridors. Classrooms added shell and spiral features to the space through the paintings on the walls. Sensory variability and information richness created through visual sense (natural colors or toys), auditory sense (music), and tactile sense of touch (natural materials) were the common features in all of the spaces. Integration of parts into wholes could be seen through the ceiling, which was divided into parts that connected together to form the whole, and bounded space features were found in all spaces. All of the spaces offered natural light (through windows), filtered and diffused light (blinds), light and shadow, and artificial light. The classrooms, cafeteria, and multipurpose hall offered spaciousness and a sense of openness through their large areas. The classroom spaces were divided into two distinct areas: one area used chairs and tables, and the other area used pads stuffed for sitting or activities. The spatial variability feature produced a variety of play zones, with some toys found in the multipurpose hall. Transitional spaces and linked series and chains could be found in the corridors. A stage in the multipurpose hall used for play or participation activities provided a sense of separation from the main space and a sense of connection between nature and humans through the prospect and refuge attributes. Furthermore, the stage offered a central focal point for the space. All of the spaces offered geographic and cultural connections to places and landscape features. The corridors (Figure 7) and the multipurpose hall both scored higher than the average, whereas the classrooms and cafeteria were all given lower ratings than the biophilic matrix's overall average.

#### 4.1.2. Daveen Preschool

In Daveen preschool, we analyzed six classrooms, a multipurpose hall, a cafeteria, and corridors. Direct environmental features (i.e., air, sunlight) and indirect features (i.e., color, views, and vistas) were found in all of the analyzed spaces, while water (a sink) was found in the cafeteria as an indirect feature. A lot of toys could be seen in spaces with animal representations. Except for the multipurpose hall, most of the spaces had paintings and posters with botanical motifs hanging on the walls. A lot of design elements could be found in the cafeteria (Figure 8), with multiple features like tree and columnar supports and egg, oval, and tubular forms. Sensory variability, information richness, and integration of parts into wholes were added to all spaces. Most of the spaces provided were bounded spaces, except for the cafeteria and corridors. Linked series and chains were offered for both corridors and cafeteria spaces, while transitional space features were found only in corridors. Outdoor access is one of the possibilities for enriching biophilic incorporation. For example, a courtyard layout can support other spaces by adding some natural themes and representational features through the visual connection. Natural,

filtered, and diffused light and shadow attributes are provided through the windows in all spaces together, and they are accompanied by artificial light. Furthermore, the cafeteria added light in shape and form and a central focal point feature to the space through the skylight. Spaciousness features were found in most spaces except for the cafeteria and corridors. With the exception of the multipurpose hall and corridors, most of the spaces exhibited the spatial variability feature through zones of activities. Geographic and cultural connections to place, landscape orientation, and landscape features were found in all spaces. The human–nature relationship element was not present in all spaces. Generally, the majority of Daveen preschool settings (classrooms and corridors) scored a little higher than the average for the biophilic features in the matrix. The cafeteria had the second-highest score among all other preschool spaces, whereas the multipurpose hall scored lower than the average.

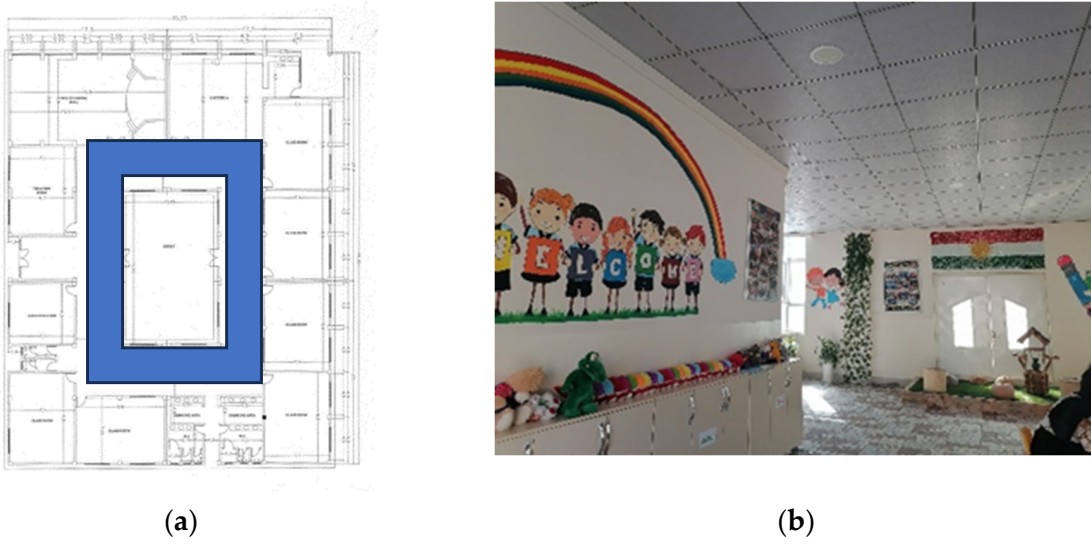

(**a**)                                                                 (**b**)

**Figure 7.** Corridors of Shang preschool: (**a**) plan drawing from the directorate of the preschool; (**b**) photo by the author.

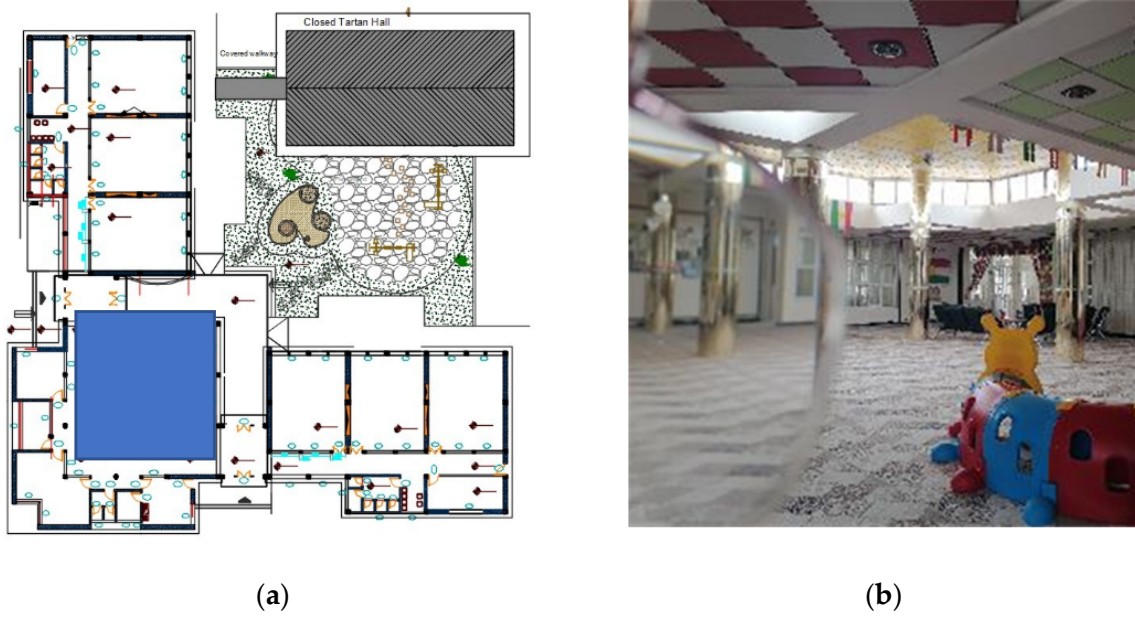

(**a**)                                                                 (**b**)

**Figure 8.** Cafeteria of Daveen preschool: (**a**) plan drawing from the directorate of the preschool; (**b**) photo by the author.

### 4.1.3. Avrocity Preschool

One of the private preschools located within an apartment complex included 11 class-rooms, a cafeteria, and a courtyard distributed over two floors. All of these spaces together displayed the highest average variance of biophilic features in the matrix. Color, water, air, sunlight, views, and vistas are examples of indirect environmental features that could be found in all spaces. Nevertheless, only the courtyard space (Figure 9) lacked water and views. A lot of photos or posters hanging on the walls could be seen in most spaces, with animal representations like fish or botanical motifs as natural shapes and forms. A variety of open and closed storage added complementary contrast features, and non-rectilinear desks or furniture in the classrooms added arches, vaults, domes, and shapes resisting straight lines. Sensory variability/information richness, integration of parts to wholes, and bounded space features were found in all spaces. Linked series and chains featured in courtyard spaces. Natural light, filtered and diffused light, and light and shadow attributes were provided through the windows in all spaces, and they were accompanied by artificial light. The courtyard added multiple features, such as spaciousness, light as shape and form, a central focal point, spatial variability, and inside/outside spaces through the skylight and the variance in the height. Geographic, ecological, and cultural connections to place features were found in all spaces. A prospect and refuge feature provided by an open small space separated from the classroom was one of the features of the human–nature relationship. Although Avrocity preschool's classrooms are generally smaller than those at Shang, Daveen, Chiman, and Kapir preschools, they are differentiated by a higher level of biophilic inclusion than those at other preschools. All of the spaces indicated were higher than the overall average of the biophilic design matrix.

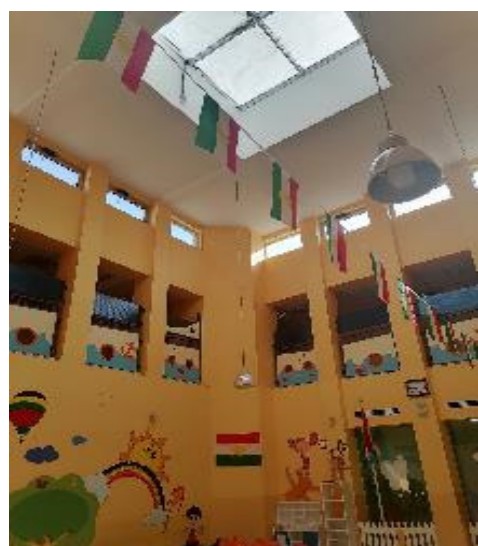 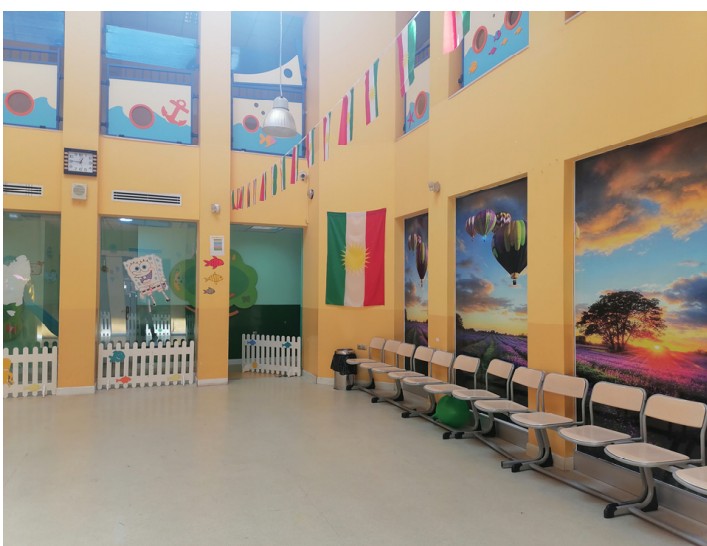

**Figure 9.** Courtyard of Avrocity preschool.

### 4.1.4. Zaryland Preschool

Another private preschool located within an apartment complex included seven classrooms, a playroom, a cafeteria, and corridors. The classrooms (Figure 10) were the smallest compared to the classrooms of all other preschools. They include chairs and tables, a whiteboard on the wall, and some open and closed storage, some of which is high and not appropriate for children to use. Storage, particularly floor-level storage that children could access, was another frequent necessity. The incorporation of direct biophilic features (i.e., sunlight, air) and indirect features (i.e., views) was added to the classrooms and cafeteria, whereas the playroom was missing these features except for the indirect feature (color), since it lacked windows. In the cafeteria, corridors, and some classrooms, wall-mounted posters, artwork, and small toys were used. These added biophilic features, including botanical motifs, animals, shells, and spirals, but they were too high and challenging for

the kids to reach, view, or touch. Sensory variability or information richness, integration of parts into wholes, and bounded space features were found in all spaces. Natural light (through windows), filtered and diffused light (blinds), and light and shadow, accompanied by artificial light features, were provided in all spaces except for the playroom, which had no windows. Geographic connection to place and cultural connection to place were found in all spaces. The lack of human–nature relationships in Zaryland preschool spaces revealed a lower level of biophilic inclusion than the overall average, especially in the playroom, which scored the worst rating for biophilia in the matrix.

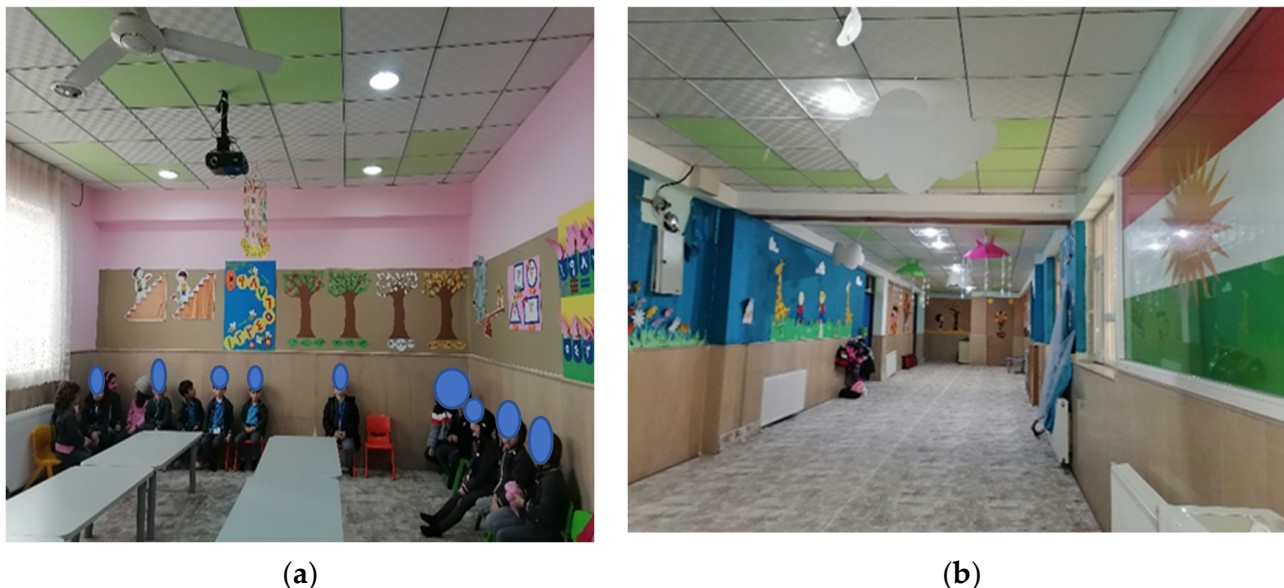

(**a**)                                              (**b**)

**Figure 10.** Zaryland preschool settings: (**a**) classroom space; (**b**) corridor space.

4.1.5. Kapir Preschool

A typical public preschool located within a residential area included five classrooms, a multipurpose hall, a cafeteria, and corridors. All of the spaces added some indirect biophilic features, such as color, water (a sink), and natural views of vegetation, as well as direct natural features like windows (sunlight) and ventilation (natural air). The presence of plant and natural material features was limited to the cafeteria and multipurpose hall. Botanical motif features were found in the classrooms, corridors, and cafeteria, while animal representations were found only in the classrooms. Sensory variability and information richness, integration of parts into wholes, and bounded space features could be found in all spaces, with corridors providing transitional space as well as link series and chains. Natural light, filtered and diffused light, and light and shadow features were provided through the windows, combined with artificial lighting in all spaces. The windows have blinds in order to moderate the light, but the height of the windows restricts children from viewing the natural views. The stage in the multipurpose hall offers a sense of separation from the main space and a sense of connection between nature and humans through the prospect and refuge attributes; furthermore, features like focal points and spaciousness in the playroom space add prospect, refuge, and special variability features to the space (Figure 11). Most of the spaces offer geographic and cultural connections to places and landscape features. The majority of Kapir preschool's settings were ranked below the average of the biophilic matrix, whereas the multipurpose hall and the cafeteria ranked slightly higher.

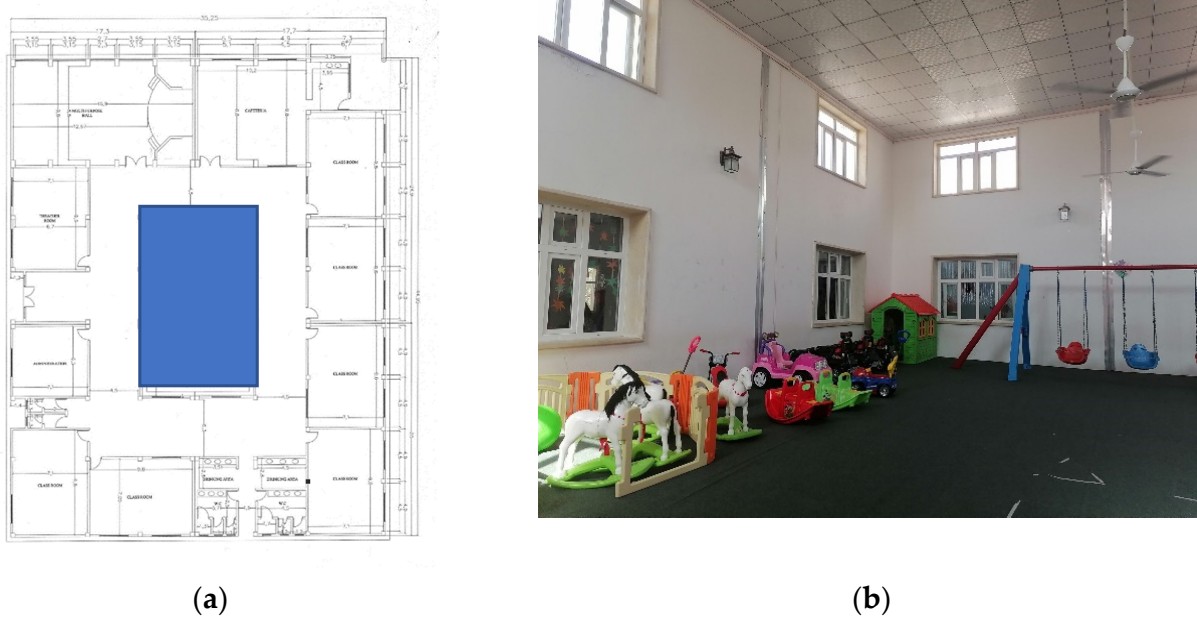

| (**a**) | (**b**) |

**Figure 11.** Play room of Kapir preschool: (**a**) plan drawing from the directorate of the preschool; (**b**) photo by the author.

### 4.1.6. Chiman Preschool

A typical public preschool located near the city center included six classrooms, a multipurpose hall, a cafeteria, and corridors. Direct environmental features (i.e., air, sunlight) and indirect features (i.e., color, views, and vistas) were observed in all spaces, while water (a sink) as an indirect feature was found in the cafeteria. Most of the spaces offered botanical motifs and animal representations through paintings and posters mounted on the walls. A few classrooms included features of shells and spirals in the space. A lot of design elements could be found in the cafeteria, with multiple features like tree and columnar supports, egg, oval, and tubular forms, and a central focal point. Sensory variability, information richness, and integration of parts into wholes were found in all spaces. Most of the spaces were provided with bounded spaces, except for the cafeteria and corridors. Links and chains were found in both the corridors and cafeteria spaces, while transitional space features were found only in corridors. Natural, filtered, and diffused light, with light and shadow attributes, is provided through the windows in all of the spaces, accompanied by artificial light. Furthermore, the cafeteria adds light as shape and form and a central focal point feature through the skylight, and attributes of both spatial variability and inside/outside spaces were found in the cafeteria as well (Figure 5). Spacious features were found in most of the spaces except for the corridors. Geographic and cultural connections to place, landscape orientation, and landscape features were found in all spaces. A lack of human–nature relationships featured in all spaces. The majority of Chiman preschool's spaces were rated lower than the average for the biophilic matrix. However, the cafeteria scored the highest among all other preschool spaces, as shown in Table 2.

The final scores of the whole matrix revealed the final averages of each preschool, which indicated Avrocity as the highest among the other preschools, as shown in Table 3.

## 5. Conclusions

The BID-M provides an innovative interior design perspective, enhancing the existing body of knowledge on biophilic design and its usefulness in the design process. It offers designers and practitioners a methodical and comprehensive vocabulary for the biophilic interior design of preschools, as well as a useful tool for incorporating and assessing biophilic features in interiors. The BID-M provides a variety of options that maximize the

presence of biophilia in the space and can support multiple biophilic features through a single attribute.

The environmental features element was the highest-scoring in some preschools, such as Avrocity, Zaryland, Kapir, and Shang, through some features such as color, which could be seen in all spaces. This element is classified as the organic dimension, which offers direct features (e.g., air, sunlight) and indirect features (e.g., views and vistas), which can be seen in most spaces. On the other hand, the light and space element scored the highest in the Daveen and Chiman preschools, through natural light, filtered and diffused light, light and shadow, spaciousness, and spatial variability, which were found in most of the spaces.

According to the matrix results, Avrocity preschool recorded a higher inclusion of biophilic features through environmental features (e.g., color, water, air, sunlight, and views and vistas), natural shapes and forms, and natural patterns and processes in comparison with the other preschools, while Daveen preschool recorded a higher inclusion of biophilic features through light and space and place-based relationships. On the other hand, Zaryland preschool recorded a lower inclusion of biophilic features through environmental features, natural patterns and processes, light and space, and place-based relationships in comparison with the other preschools. In addition, Kapir preschool recorded a lower inclusion of biophilic features through natural shapes and forms, light and space, and place-based relationships.

The matrix revealed the levels of biophilic variability in the different spaces of each preschool, where the cafeteria at Chiman preschool showed the highest incorporation of biophilic features, followed by the cafeteria in Davin preschool and the classrooms in Avrocity preschool.

Remarkably, one of the most common issues discovered in most of the preschool spaces studied—especially in the Daveen and Chiman preschools—was a lack of biophilic features that strengthen the bonds between nature and humans, referred to as the human–nature relationships element, although this was minimally observed in Avrocity preschool. It can be supported by some small formations, such as independent play places that have privacy from the overall space and overlook the larger space in which they are located. Prospect and refuge refer to places where children enjoy playing when they feel they are in a special space. Another opportunity comes through the element of temptation and curiosity, perhaps through the creation of some features such as lighting and shadow or light and dark, and the presence of elements that stimulate the child's curiosity to discover order and complexity through furniture, materials, or colors.

The presence of some places with limited areas and the absence of windows resulted in a lack of daylight and natural air; thus, those spaces received the lowest ratings for biophilic design attributes. There are opportunities to make areas more biophilic by expanding them and adding some windows. Furthermore, the incorporation of biophilia through place-based relationships can be supported by outdoor access, which connects indoor spaces to the surrounding landscape.

Overall, some common issues were observed in these interiors. For example, the storage used by children in some spaces was high and difficult to reach. It would be better to design this furniture with consideration of children's scale and the variations of open or closed storage to make its use easier. Also, the posters and photos mounted on the walls are another issue to be considered when children cannot reach them to see or touch them for more interaction with these natural paintings. Additionally, the window height from the ground floor in some cases restricted children's ability to enjoy the natural views behind the windows, such as butterflies or feeling the breeze. Including natural themes and representations of the natural world in the spaces (e.g., Sun, plants, water, animals, or color) would make them more biophilic. Adding natural materials rather than artificial ones would be more biophilic and healthier. Another significant issue that was observed throughout the preschool environments was the unintentional random use of colors.

## 6. Further Study

Implications throughout time: Using biophilic design elements in preschool interiors may have long-term implications, which could be investigated in future studies. Researchers can measure the long-term effects on children's wellbeing, cognitive development, and academic achievement through longitudinal studies, offering important insights into the long-term advantages of biophilic environments.

Design interventions: More research is required to determine the efficacy of particular design solutions intended to improve biophilic features in preschool spaces. Researchers can investigate the use of naturally inspired components, including living walls, organic materials, and daylighting techniques, and evaluate their effects on students' engagement, concentration, and overall learning results.

Further research is needed to explore the health and wellbeing benefits of biophilic design in preschool environments. Factors such as stress reduction, immune system performance, creativity, and emotional wellbeing could be investigated. Cultural context and diversity are crucial for ensuring inclusion and meeting diverse community needs. Future studies should explore how cultural backgrounds, geographic regions, and socioeconomic issues affect the integration and effectiveness of biophilic features.

Understanding how users—children, instructors, and parents—perceive and behave in relation to biophilic design in preschool spaces is another potential research subject. Insights into user preferences, satisfaction, and the possibility for behavioral changes can be gained via qualitative research, surveys, and interviews about how various stakeholders interpret, value, and use biophilic features.

Researchers can improve their knowledge of biophilic design in preschool interiors by delving into these areas for future research, which will ultimately help to develop evidence-based recommendations and tactics for making learning environments for young children that are healthier, more interesting, and connected to nature.

**Author Contributions:** Conceptualization, I.M.; methodology, I.M.; software, I.M.; validation, I.M.; formal analysis, I.M.; investigation, I.M.; resources, I.M.; data curation, I.M.; writing—original draft preparation, I.M.; writing—review and editing, I.M.; visualization, I.M.; supervision, Z.O. and Ç.Ç. All authors have read and agreed to the published version of the manuscript.

**Funding:** This research received no external funding.

**Institutional Review Board Statement:** The study was conducted in accordance with the Declaration of Helsinki, and approved by the Institutional Review Board of the Directorate of Duhok Education (20 October 2022). All authors have read and agreed to the published version of the manuscript.

**Informed Consent Statement:** Informed consent was obtained from all subjects involved in the study.

**Data Availability Statement:** As stated in the full article, all data is accessible publicly.

**Conflicts of Interest:** The authors declare no conflict of interest.

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
