# Peer review of "An Exploration of Biophilic Design Features within Preschool Interiors"

_sustainability, doi:10.3390/su151511913_

Round 1
Reviewer 1 Report
1. Authors need to enhance the abstract.
2.Explanation needed for biophilic content.
3. Introduction part of the study was very small. I kindly request the authors to give large introduction.
4. Protocol design and explanation was not sufficient.
5. if any mathematical models for BIDM? please explain it.
6. More explanation needed in terms of comparison results for pre schools.
7. Latest references needed.
8. Future work of the study was not explained properly.
Author Response
Dear reviwer1
Here I sent the Word files
1-Response to reviewer1 comments
2- Whole manuscript modifications

Reviewer 2 Report
The title of the article announces an interesting research, I have however, some suggestions to improve the quality of the paper:
Abstract: consider merging the information in lines 13-16 into a single sentence
Introduction: please emphasize how this research is new and innovative and not just an analysis of biophilic features in schools and an application of BIDM (which would not be at the level of a research paper). Is this matrix new? Also please highlight the research questions of the article.
Literature review: Please explain what the conclusion of this Section is, and how your paper will fill the research gaps, how your paper will go beyond the level of research so far. Literature review should analyze,synthesize, and critically evaluate to give a clear picture of the state of knowledge on the subject. In your article it's just a list of papers.
Please explain what Measures - line 205 means. I suggest you replace it with a more appropriate expression for the Materials and methods section. Also in this section you could summarize table 1, maybe only with the 6 features.
The Results and Discussion section informs about the qualitative survey results. What survey are we talking about? Why isn't the content and information about the survey introduced under Materials and Methods? Also here you might as well explain the novelty of the approach because in the results you talk about the scores, which could be just an applied part of a research but not a research per se.
Consider enriching the bibliography, especially as you have a Literature review section.
Small corrections are necessary:
In Line 187: To assess the biophilic content to support children's health.
Acronyms should be explained first time when they appear in the text: I assume RED in line 76 means Restorative environmental design in line 82. Similarly what does WELL means in line 149?
In line 88, it would be appropriate to include Award et al.
Please explain how many attributes are considered: 52 in line 44 or 53 in line 194? How many attributes are there in total and why can't they all be considered? In line 196 you refer to the remaining 72 and in line 144 to the original 72 attributes.
In the end, please increase the clarity of the paper, to view the red line between the different sections of the paper.
Author Response
Dear reviewer 2
Here I sent the Word files:
1- Response to reviewer 2 comments
2- Whole manuscript modifications

Round 2
Reviewer 2 Report
The authors have responded to the reviewer's comments. I would however, kindly ask the authors to respect the condition on 200 words abstract. Afterwards, in my opinion, the paper may be published. Also, further study may be included as a paragraph in the Conclusion section.
Author Response
Dear Reviewer:
Here, you can find the attached file related to the response to the reviewers comments.
